# Sun-Protective Clothing Worn Regularly during Early Childhood Reduces the Number of New Melanocytic Nevi: The North Queensland Sun-Safe Clothing Cluster Randomized Controlled Trial

**DOI:** 10.3390/cancers15061762

**Published:** 2023-03-14

**Authors:** Simone L. Harrison, Petra G. Buettner, Madeleine J. Nowak

**Affiliations:** 1Skin Cancer Research Unit, College of Public Health, Medical and Veterinary Sciences, James Cook University, Douglas, Townsville, QLD 4811, Australia; 2Australian Institute of Tropical Health and Medicine (AITHM), Townsville, QLD 4811, Australia; 3Australian Institute of Tropical Health and Medicine (AITHM), Cairns, QLD 4875, Australia; 4College of Medicine and Dentistry, James Cook University, Douglas, Townsville, QLD 4811, Australia

**Keywords:** clothing, sun protection, melanocytic nevus, ultraviolet protection factor (UPF), melanoma, skin cancer, garment, ultraviolet radiation, dermatology, children

## Abstract

**Simple Summary:**

The number of pigmented moles is the most important risk factor for melanoma. This “real world” study conducted in 25 Australian childcare centers is the first to show it is possible to prevent a significant proportion of pigmented moles in young Caucasian children by dressing them in UPF 30-50+ clothing that covers at least half their body on a daily basis. Regularly wearing this clothing for 3.5 years was sufficient to prevent almost one-quarter of the pigmented moles that young children developed on their skin. This should, by implication, reduce their risk of developing melanoma in the future. Primary health care providers should stress the importance of garment coverage when recommending sun protection for children and should lobby for all sun-protective clothing standards to assess and report garment coverage, in addition to the transmission of ultraviolet radiation through the fabric (reported as UPF), on clothing labels and packaging for children’s clothing, including school uniforms.

**Abstract:**

Numerous pigmented moles are associated with sun exposure and melanomarisk. This cluster randomized controlled trial aimed to determine if sun-protective clothing could prevent a significant proportion of the moles developing in young children (ACTRN12617000621314; Australian New Zealand Clinical Trials Registry. Twenty-five childcare centers in Townsville (19.25° S), Australia, were matched on shade provision and socioeconomic status. One center from each pair was randomized to the intervention arm and the other to the control arm. Children at 13 intervention centers wore study garments and legionnaire hats at childcare and received sun-protective swimwear and hats for home use, while children at the 12 control centers did not. The 1–35-month-old children (334 intervention; 210 control) were examined for moles at baseline (1999–2002) and were re-examined annually for up to 4 years. Both groups were similar at baseline. Children at intervention centers acquired fewer new moles overall (median 12.5 versus 16, *p* = 0.02; 0.46 versus 0.68 moles/month, *p* = 0.001) and fewer new moles on clothing-protected skin (6 vs. 8; *p* = 0.021 adjusted for confounding and cluster sampling) than controls. Intervention children had 24.3% fewer new moles overall (26.5 versus 35) and 31.6% (13 versus 19) fewer moles on clothing-protected skin than controls after 3.5 years. Sunlight’s influence on nevogenesis is mitigated when children regularly wear UPF 30-50+ clothing covering half their body, implying that increased clothing cover reduces melanoma risk. Sun-protective clothing standards should mandate reporting of the percentage of garment coverage for childrenswear.

## 1. Introduction

Queensland has the highest reported incidence of keratinocyte skin cancers [1] (including basal and squamous cell carcinomas) and cutaneous melanoma globally [2], and overexposure to solar ultraviolet radiation (UVR) is the main environmental cause of both neoplasms [3,4,5,6]. Most cases of cutaneous melanoma in Australia, New Zealand, North America, and Europe can be prevented by avoiding excessive UVR exposure—the only modifiable risk factor for cutaneous melanoma [6]. Although Australians have been exposed to forty years of skin cancer prevention campaigns encouraging them to adopt personal sun protection practices [7], the most recent statewide survey conducted in Queensland in 2020 revealed that 49 percent of adults and 45 percent of children had been sunburnt during the previous year, with about 33,000 of these children experiencing five or more episodes of sunburn in 12 months [8]. The population of tropical North Queensland (NQ) is exposed to high to extreme levels of solar UVR year round [9], and regional dosimetry studies of children and adults suggest they receive a high UVR dose in both summer and winter [10,11,12].

Reports of cutaneous melanoma arising in a pre-existing melanocytic nevus (pigmented mole) vary widely from 4 percent to 72 percent [13]. However, recent advances in imaging technology and molecular biology and the application of prospective study designs suggest that melanocytic nevi (MN) are precursors in about 30 percent of cases of cutaneous melanoma [13,14,15]. The likelihood of developing melanoma before age 30 is also greatest when there is clinical evidence of early-acquired or large MN [16]. Numerous MN is the major phenotypic risk factor for cutaneous melanoma [5,17,18,19,20,21], with the presence of atypical [17,20] and large MN [16,20] further increasing risk. The number of MN reflects both genetic factors (accounting for about 58% of the variability in MN counts [22]) and UVR exposure [23,24].

Infants and children are particularly vulnerable to the effects of excessive sun exposure [24,25,26,27,28,29,30,31], and UVR-related changes in an infant’s skin have been shown to begin accumulating from their first summer [31]. Migrants relocating to UVR-intense climates after ten years of age develop fewer MN and are less likely to develop melanoma as adults than native-born residents or migrants who arrive earlier in life [24,32], suggesting that childhood is a critical time for the influence of sun exposure in MN development [24,25,26,27,28,29,30,32,33,34,35]. Children raised in NQ acquire common MN, large MN, and atypical MN earlier and in higher numbers (up to 252 common MN by age six) than children raised in less intense solar UVR environments [27,28,29,30,33], putting them at increased risk of developing melanoma in the future [5,17,19,20,21,24,32]. Multivariate results from our earlier Townsville (NQ) preschool cohort study showed that the total number of hours children spent in the sun between annual skin examinations and their tendency to burn on exposure to sunlight were independent predictors of the rate at which they developed new (incident) MN [29]. Stratification of these results showed that tendency to burn was an effect modifier. The relationship between total body incident MN count and the number of sunburns experienced in the year between skin examinations by these two- to seven-year-old Caucasian children was only significant in children who had a low to moderate sunburn tendency, but not those who were very sun sensitive. Total number of sunburns experienced by these young children since birth was, however, an independent predictor of the presence of MN that were at least five millimeters in diameter by their annual follow-up examination [29]. More recently, our 20+-year follow-up of this pediatric cohort found that the rate of acquisition of new MN between early childhood and the third decade of life was significantly higher in individuals who had experienced a sunburn before seven years of age, particularly if the sunburn was severe or had occurred annually during early childhood [35]. Other recent epidemiological studies of MN [26,34], including a cross-sectional analysis of baseline data from the “Study of Nevi in Children” (SONIC) by Satagopan and co-workers [34], corroborate the findings of earlier studies of MN, demonstrating the importance of sunburn in early life and host factors such as sun sensitivity in their development [24,28,29]. These findings closely resemble those of a large meta-analysis showing that a single episode of sunburn during childhood is sufficient to almost double the likelihood of developing melanoma in the future [36]. Recent genomic studies have also improved our understanding of the complexities of nevogenesis, including the role of UVR exposure in this process. Stark and co-workers [37] found UVR mutation signatures similar to those observed in melanoma in 97% of the acquired MN they examined, but only 10 percent of matched peri-lesional skin samples. A recent meta-analysis of 11 genome-wide association studies of MN in 52,506 individuals produced results suggestive of multiple pathways in nevogenesis [22]. This is consistent with the findings of a recent comprehensive review of the biology and genetics of MN that concluded that their anatomical distribution and signature mutations support the existence of UVR-related and non-UVR pathways in the development of MN [38]. Similar conclusions have been reached by dermoscopic [18] and epidemiological studies [39,40,41] that compared MN by age, anatomical site, and inferred pattern of sun exposure. It is anticipated that future research that increases our understanding of nevogenesis and the natural history of MN will further elucidate the biological pathways to melanoma development, but this remains a work in progress [38,42,43,44].

Only about two to three percent of Caucasian babies are born with any MN [30]. Caucasian children raised in NQ begin acquiring benign MN earlier (6–12 months old) than ethnically similar children from Scotland (12–36 months) [30]. The rapid acquisition of MN in NQ continues until about 12 years of age, when the rate of development of MN begins to plateau so that, by 15 years of age, adolescents from Melbourne (latitude 37.8° S), Sydney (34.0° S), and Townsville (19.25° S) have similar MN counts [27]. This inverse association between latitude of residence and MN frequency in prepubescent Caucasian Australian children [27] reflects the latitudinal gradient in cutaneous melanoma incidence between these cities [2]. Comparisons of MN counts in countries with contrasting melanoma rates and ambient UVR levels produce similar results. For example, the incidence of cutaneous melanoma in Australia is much higher than in the United Kingdom [2,45], and MN are more prevalent in young Australian children than in their British counterparts [30]. However, MN frequency is similar in older individuals from these contrasting climates, suggesting that early acquisition of MN increases melanoma risk [16,30,45]. Because sun exposure is related to MN development during childhood [23,24,25,26,27,28,29,30,32,35,36], and MN count and sun exposure are both risk factors for cutaneous melanoma [3,4,5,6,13,14,15,16,17,18,19,20,21,22,24,32,39,40,43,45], MN counts provide a short-term measure of the efficacy of interventions designed to reduce sun exposure and subsequent melanoma risk [46,47,48]. Australia is recognized internationally for its skin cancer prevention programs dating back to the 1980s [7]; thus, when our randomized controlled trial (RCT) commenced, most (97.8 percent) accredited childcare centers in Queensland already had a written sun protection policy requiring children to use high sun protection factor (SPF) sunscreen and wear a hat at childcare [49], while protective clothing was underutilized [46]. Moreover, when we were planning our RCT, most of the studies showing promise for preventing MN pointed to sun-protective clothing [50,51], which, unlike sunscreen, provides uniform broad-spectrum UVR protection without the pitfalls of application frequency, uniformity, and thickness [52,53]. Consequently, our intervention study aimed to determine whether regularly wearing sun-protective clothing that covers a considerable proportion of the body’s surface area (BSA) during early childhood reduces sun exposure sufficiently to prevent the development of 24 percent of MN by four years of age, making total MN counts for Townsville children similar to those expected for children from the less UV-intense climate of Sydney [27,28,29,30,33]. It is well understood that many though not all MN develop in response to DNA damage caused by UVR [15,22,23,38,39]. Consequently, not all MN can be prevented. To our knowledge, this is the first prospective study to measure the efficacy of clothing in preventing MN. Our findings highlight the importance of increased clothing coverage as a strategy to reduce the number of MN acquired during childhood and, by implication, future risk of cutaneous melanoma. These findings are likely to influence the direction of skin cancer prevention strategies targeting children, particularly the standards currently governing the marketing and labeling of sun-protective clothing.

## 2. Materials and Methods

### 2.1. Study Setting and Rationale

This study was conducted in Townsville (latitude 19.25 °S), a regional city situated on the NQ coast of Australia, which has a dry tropical climate and experiences high to extreme ambient UVR levels year round [9]. Ethics approval was provided by the James Cook University Human Ethics Committee (Approval Number: H804, 15 October 1999) to conduct a cluster-RCT (c-RCT) in children under three years old who regularly attended a participating childcare center in Townsville. Our intervention study aimed to determine the efficacy of sun-protective clothing in preventing the development of MN during early childhood. More specifically, this “real-world study” sought to determine if regularly wearing clothing rated as providing “very good” to “excellent” UVR protection [54] that fully covered the trunk, upper arms, thighs, and posterior neck until 4+ years old (prior to starting school) could reduce MN counts by 24 percent in a cohort of 1–35-month-old Caucasian children attending childcare centers (clusters) randomized to the clothing intervention in comparison to a cohort of children the same age at control childcare centers (clusters). This aim was based on reducing the median total body MN count for 4-year-old Townsville children to that expected for children the same age from the lower-solar irradiance environment of Sydney, Australia [9], and required us to recruit and retain approximately 230 children in each of the two study arms until they had participated in follow-ups for at least 3 years. Because the Townsville population can be quite transient, recruitment was inflated to account for loss in terms of incomplete follow-ups. Power calculations for this c-RCT were previously published [46].

### 2.2. Recruitment of Childcare Centers

Prior to commencing the c-RCT, all 28 of the licenced childcare centers operating in Townsville in June 1999 were invited to participate in a shade audit, as described elsewhere [46]. During the shade audit (100 percent response), it became apparent that 2 of the 28 centers did not enrol infants. To be eligible to participate in the c-RCT, childcare centers (clusters) had to meet the inclusion criteria of being an accredited long daycare center that operated 5 full days/week and provided care for infants in a separate baby unit(s). The remaining 26 eligible centers were paired on shade provision (three levels) and the socioeconomic index for areas (SEIFA) for the suburb in which they were located (three levels) [55]. One childcare center from each pair was then randomly allocated to the intervention arm and the other to the control arm by the statistician (P.G.B.).

The trial was registered as ACTRN12617000621314 in the Australian New Zealand Clinical Trials Registry (https://www.anzctr.org.au, accessed on 7 March 2023) [56], and the study design, detailed methods, and baseline results have been published elsewhere [46]. Briefly, the directors of the 26 eligible childcare centers were invited to participate in the clothing c-RCT by S.L.H. during the shade audit feedback session. Twenty-five childcare centers consented to being involved (Figure 1).

### 2.3. Recruitment of Children

A list providing the first name, date of birth, and attendance pattern of all enrolled children was obtained from the director of each of the 25 participating childcare centers. The parents/carer(s) of all children aged 1–35 months who were enrolled to regularly attend a participating childcare center received a study information sheet, a baseline questionnaire, and a consent form via the childcare center that invited their child to participate. Children were recruited into the study from November 1999 until July 2002 to achieve an adequate sample size. The director of each participating childcare center regularly provided updated enrollment and attendance lists to help us accomplish this aim.

### 2.4. Questionnaires

Consenting parents were asked to complete a self-administered baseline questionnaire on behalf of their child prior to the baseline skin examination for MN. The parental questionnaire sought information about each participating child’s ethnicity, residental history, demographic characteristics, sensitivity to sunlight, use of protective clothing and sunscreen (including a diagram of the body parents shaded to show where they usually applied sunscreen on their child), and sunburn history by indicating how often their child had experienced each of the three types of sunburn (redness/peeling/blistering sunburn) and shaded corresponding areas of a diagram of the body to show where they had been sunburnt [46]. Parents indicated the time their child typically spent in the sun between 6 a.m. and 7 p.m. separately for weekdays and weekend days. Parents were also asked how often during the warmer half of the year their child played outdoors and engaged in other outdoor activities such as swimming.

The frequency of the three types of sunburn were weighted 1, 2, and 3 and then summed to produce a sunburn severity score, while extent of sunburn was determined by summing the BSA of affected body sites [58,59]. These two scores were multiplied to produce a combined extent and severity of sunburn score. Likewise, a sunscreen score was created by summing the BSA of body sites where sunscreen was applied [58,59] and multiplying by frequency of sunscreen use at home during (1) summer and (2) winter. A hat use score was calculated for each child by multiplying the hat style they wore most often at home (no hat = 0, cap = 1, wide-brimmed = 2, legionnaire = 3) by the frequency of use summed for (1) summer and (2) winter. Similarly, a swimwear score was created by multiplying usual swimwear style (naked = 0; shorts/briefs = 1; bikini = 2; full-piece swimsuit = 3; shorts/full-piece/briefs plus a T-shirt = 4; cover-up Lycra swimsuit (nylon elastane; E.I. du Pont Nemours & Co., Inc., Wilmington, DE, USA) = 5) by frequency of use during water activities summed for both summer and winter. Additionally, a senior staff member from each baby/toddler unit (n = 76 units) within the 25 participating childcare centers completed a survey about children’s patterns of outdoor exposure and sun protection practices soon after the intervention study began.

### 2.5. Clinical Examination of Children

Children whose parents/carer(s) provided written informed consent for them to participate were examined for MN at baseline and annually thereafter for up to four years (whilst the child attended the childcare center). S.L.H. conducted full-body skin examinations at 30 body sites (excluding the buttocks, genitals, and scalp) on all intervention and control children to enumerate MN of all sizes using a standard international protocol [60]. It was impractical to blind participants, parents, and carers in the intervention arm to the random assignment of their childcare center because study garments were provided to intervention childcare centers and sun-protective swimwear and hats were provided for children to wear at home. Likewise, it was not feasible to blind the examiner (S.L.H.) because of her close involvement in implementing the c-RCT. However, very high concordance was found for MN counts performed on 49 children by S.L.H. and an experienced visiting dermatologist who was blinded to assignment groups (concordance correlation co-efficient 0.97, 95% confidence interval (95% CI) 0.95, 0.98) [46].

Height and weight were recorded for each child, together with their, skin, hair and eye color. Skin reflectance for the sun-protected surface of the inner upper arm was also measured three times at 685 nm using a reflectance spectrophotometer (Evans Electroselenium Ltd., model 99; Diffusion Systems Ltd., London, UK) (high skin reflectance values indicate fair skin). Average reflectance was then categorized as either fair, medium, or olive using established cut points [27,28]. The methods applied to each study group are summarized in Appendix A.

### 2.6. Intervention Design

We provided the 13 childcare centers in the intervention arm with sufficient UPF 50+ legionnaire hats, elbow-length cotton crew neck T-shirts with UPF values ranging from 32.1 to 44.9 (Appendix A), and UPF 40 rated (blocks 97.5 percent of erythemally effective UVR [54]) water-resistant nylon taslon knee-length shorts to clothe all children enrolled at these centers. Intervention childcare centers were also provided with long-sleeve UPF 40 taslon shirts for the children to wear during water-based outdoor activities. These garments were manufactured exclusively for the study, and they were not available to purchase until after the study concluded to prevent contaminating the control arm. Staff at intervention childcare centers were trained to ensure all children under 3 years old were dressed in study garments that fully covered the trunk, upper arms, and thighs on arrival at the center each day and they were changed back into their own clothing just prior to leaving. Study T-shirts and shorts were color-coded by size to assist staff when dressing children. Signs were placed on the doors of intervention childcare centers that led to outdoor play areas to remind to staff to ensure all children were dressed in study garments before heading outside.

As the eldest study participant at each intervention childcare center moved up to a higher age group/unit, we supplied the new unit with enough appropriately sized study garments to outfit all children. We also trained staff in the new unit to ensure that the study garments they dressed the children in fully covered the thighs to below the knees and the arms to below the elbows. Study garments were worn by children at all 13 childcare centers in the intervention arm for 5.5 years until July 2005, when the last child had their third annual skin examination.

A c-RCT design was used because it was the simplest to implement. Allocating the intervention by cluster (childcare center) meant staff at the 13 intervention childcare centers dressed all children in study garments each day, whereas randomizing by individual would have required staff members at all 25 childcare centers to remember which children to dress in study garments and which not to. Although eligible children were those who met the inclusion criteria of being less than 36 months old at their baseline skin examination, having two or more grandparents of European origin, being enrolled to regularly attend a participating childcare center, and being likely to remain in Townsville for the foreseeable future, all children at intervention childcare centers wore study garments and benefited from the intervention, irrespective of their eligibility. Data were only collected from children whose parents provided written informed consent. Children with fewer than two grandparents of European descent took part in the intervention but were excluded from the dataset prior to statistical analysis because they represented a much lower-risk group.

### 2.7. Compliance

The proportion of children observed wearing study garments at intervention centers was recorded twice a week when the research assistant attended to collect dirty laundry and replenish the supply of clean study garments. Weekly laundry volumes were also monitored. These safeguards enabled us to intervene if compliance appeared to be dwindling. Parents of eligible children attending intervention childcare centers were also educated about the correct use of study garments and, prior to the start of summer each year, were given UPF 50+ protective swimwear and a legionnaire hat for their child to wear at home.

### 2.8. Control Childcare Centers and Children

Children attending childcare centers assigned to the control arm of our c-RCT wore their own clothing at childcare (they did not receive study garments) and staff at these centers were advised to continue with their center’s normal sun protection practices. Children at control childcare centers were examined annually for MN at baseline and annually for up to four years, and their parents/carer(s) completed the same questionnaires as the parents/carer(s) of the children attending intervention centers. Any movement of children and/or staff between intervention and control childcare centers was also monitored.

The Cancer Council Queensland delayed implementing their SunSmart Early Childhood Program [61] in Townsville after we notified them of our study to avoid contaminating the results. Consequently, only one intervention and one control center were enrolled in the SunSmart Early Childhood Program when our c-RCT commenced, and interactions between these two childcare centers and the Cancer Council were put on hold until after our c-RCT ended.

### 2.9. Statistical Analysis

Categorical data were described as percentages. Depending on the distribution, numerical data were summarized as mean and standard deviation (SD) or median and interquartile range (IQR). Comparisons between childcare centers (clusters) were performed using Chi-square tests and non-parametric Wilcoxon tests. All 25 childcare centers participated for the entire duration of the study. Baseline comparisons between children attending intervention and control centers were conducted using Chi-square tests, non-parametric Wilcoxon tests, and t-tests, as were baseline comparisons for children who were and were not subsequently lost to follow-up. Analyses were based on original assigned groups. Comparisons of sun-related behaviors within childcare center units and of children’s baseline characteristics were adjusted for the cluster effects of the childcare centers. For these cluster-adjusted comparisons, skewed numerical data were logarithmically transformed, resulting in approximately normal distributions. Outcome measures were reported by cluster and included the total number of new MN at follow-up (i.e., baseline whole-body MN counts subtracted from whole-body MN counts at the final follow-up examination for each child); the number of new MN acquired per month of follow-up (incidence rate); and the number of new MN on body sites covered by “study garments” (sum of new MN on the trunk, upper arms, thighs, and posterior neck) at follow-up. At the final follow-up, all three outcome measures were compared for the intervention and control arms using (1) non-parametric Wilcoxon tests, (2) cluster-adjusted Wald tests comparing the mean values of logarithmically transformed measures, and (3) cluster-adjusted multiple regression analyses of logarithmically transformed measures to adjust for confounding variables. A separate analysis was performed for each of the three different outcome variables. After a stable model was achieved for the logarithmically transformed target variable of interest using a stepwise selection approach, the variables remaining in the model were considered as potential confounders. A characteristic was considered a confounder when it changed the estimated coefficient of the original regression model by more than 5 percent, and the model was adjusted accordingly. All possible interactions between variables of one model were defined and checked for significance in hierarchical models. In addition, 95% CIs were calculated for the differences between median incidences of MN in the intervention and control groups and were adjusted for the cluster sampling approach (STATA cendif command). The level of compliance of intervention centers was based on observational data collected during the 1885 occasions project staff were at an intervention center replenishing the supply of laundered garments while children in the study were outdoors in the sun. Level of compliance was determined for each of the 13 intervention centers by calculating the mean and median proportion of children observed wearing study garments outdoors from all available data. The average proportion of children observed wearing study garments outdoors was used to rank clothing compliance at the 13 intervention centers from highest average percentage compliance to lowest average percentage compliance and create an ordinal variable with 13 categories. The non-parametric Jonckheere–Terpstra (J-T) test was then used to test for a linear trend in the total number of incident (new) MN (continous variable) developing in the 334 children in the intervention group across the 13 levels of compliance (performed in IBM SPSS version 29). Following a non-significant result, the 13 ordinal clothing compliance categories were collapsed into fewer categories and the J-T test was repeated to test for an ordered alternative hypothesis (i.e., that median incident MN counts increase as clothing compliance decreases) within a smaller number of independent subgroups. In instances where the J-T test produced an asymptotic 2-tailed *p*-value < 0.05, IBM SPSS version 29 automatically performed Mann–Whitney U test post hoc pairwise comparisons (i.e., for two levels of compliance) for all possible combinations. A one-tailed *p*-value was provided for each pair of tests after being adjusted for multiple comparisons using Bonferroni’s correction (alpha = 0.05/number of comparisons, e.g., 5 categories of compliance, alpha = 0.05/10; therefore, *p* < 0.005 required to reach statistical significance).

All statistical analyses, except for the J-T test pertaining to differences in incident MN counts by level of clothing compliance, were performed by P.G.B. using SPSS for Windows version 22 and STATA release 12.0 (Stata Corporation, College Station, TX, USA). J-T tests were performed by S.L.H using IBM SPSS for Windows version 29. We followed the consolidated standards of reporting trials (CONSORT) [57], and the CONSORT checklist is provided in Appendix A.

## 3. Results

In total, 25 of the 26 eligible childcare centers consented to participate in the clothing c-RCT (96.2% response). One childcare center declined to participate due to uncertainty following the recent death of their owner–director. Consequently, the c-RCT included 12 control childcare centers (92.3% response) and 13 intervention childcare centers (100% response, Figure 1).

A similar proportion of intervention and control childcare centers were commercial (privately owned) services (61.5% versus 58.3%; Table 1), while the remainder were community-based centers. Each participating childcare center operated two to four baby/toddler units for children under 3 years old. There were no significant differences in management characteristics and sun protection policies between intervention and control childcare centers, nor were there any differences in sun exposure, sunscreen use, or shade quality between intervention and control baby/toddler units (Table 1).

A total of 1136 children were cluster-randomized during the 2.5-year recruitment period (Figure 1). Of the 616 children randomized to the intervention arm, 144 were ineligible (8 were older than 35 months old when examined for MN; 10 had fewer than two grandparents of European origin; 114 did not intend to remain in Townsville; and 12 attended childcare infrequently) and 33 declined to participate (Figure 1). Consequently, baseline skin examination and questionnaire data were obtained for 439 of the 472 eligible children from intervention centers (93% response). Similarly, 129 of the 520 children attending control childcare centers were ineligible (14 were too old; 8 were non-Caucasian; 105 were leaving Townsville; and 2 attended childcare infrequently) while 81 declined to participate (response 79.3%). The proportion of eligible children that had a baseline skin examination was 86.8% (n = 749), 544 of whom (63%) had at least one follow-up examination. Loss to follow-up was 23.9% in the intervention arm (left Townsville n = 46; left center n = 53; stopped wearing clothing n = 3; infrequent attendance n = 3) and 32.3% in the control arm (left Townsville n = 26; left center n = 74). No evidence of harm or unintended effects from the intervention was identified.

Eligible children whose parents provided written informed consent to participate and who were examined at baseline and at least one follow-up examination at a participating childcare center were included in the study cohort (n = 544; Figure 1). There were no marked differences at baseline between the 334 children in the intervention arm and the 210 children in the control arm, except that none of the children in the intervention arm had experienced a blistering sunburn, while three controls (1.5%) had (*p* = 0.022). Despite this, no significant differences were evident for the extent or severity of sunburn or the composite sunburn score (Table 2). Although initially the baseline hat use scores appeared to be slightly different for intervention and control centers, the difference was not statistically significantly after adjustment for cluster sampling (*p* = 0.143). Similarly, the median total number of MN present at baseline on intervention and control children was not significantly different when the mean log-transformed total MN count was compared for the two groups after adjusting for cluster sampling (*p* = 0.183; Table 2).

The 205 children with a baseline examination who were lost to follow-up were not markedly different from the 544 children with follow-up data. Comparisons between the two groups with respect to all characteristics presented in Table 2 only resulted in two statistically significant findings. Children who were lost to follow-up were less likely to have fair skin based on a subjective assessment by the examiner (89.0% vs. 97.2%; *p* < 0.001) and were less likely to have been born in the tropics (86.2% vs. 91.7%; *p* = 0.018, data missing for n = 107) than children who completed the study.

Of the 544 children who had follow-ups, 151 (27.8%) had follow-ups for less than 18 months, 347 (63.8%) had follow-ups for 18–41 months, and 46 (8.5%) had follow-ups for ≥3.5 years (Table 3). The median number of new MN at follow-up was significantly higher in the control group (16; IQR = [8, 30]; range 0–77) than the intervention group (12.5; IQR = [5.75, 23]; range 0–74: *p* = 0.020) overall (adjusted for confounding and cluster sampling) and for each year of follow-up (Table 3). The cluster adjusted 95% CI for the overall median difference in new MN at follow-up ranged from −7 to −1.

The median number of new MN acquired per month of follow-up was significantly higher in the control group (0.68 MN/month; IQR = [0.37, 1.04]) than the intervention group (0.46 MN/month; IQR = [0.25, 0.71]: *p* = 0.001; adjusted for confounding and cluster sampling; 95% CI for median difference = [−0.29 to −0.07]; Table 3). MN incidence rate plateaued for intervention children who were retained in the study for longer periods, while in control children MN incidence continued to rise with the duration of follow-up. Similar MN incidence rates, 0.58 and 0.57 MN per month, were observed for intervention children who were retained in the study for 30–41 months and 3.5 years or longer, respectively, while rates of 0.75 and 0.81 MN per month were observed for controls followed for the same duration (Table 3). Consequently, the longer the period spent in the study, the greater the difference in the total number of new MN observed between the groups, such that intervention children followed for at least 3.5 years developed 24.3 percent (26.5 versus 35) fewer new MN in total than control children (Figure 2; Table 3).

The median number of new MN developing on body sites specifically protected by study garments was also significantly higher in controls (8; IQR = [4, 15.25]) than in the intervention group (6; IQR = [2, 12]: *p* = 0.021; adjusted for confounding and cluster sampling; Table 3). The cluster-adjusted 95% CI for the median difference in new MN on clothing-protected body sites was −4 to 0. The greatest difference between the groups was seen for children who were followed the longest, with intervention children followed for at least 3.5 years developing 31.6 percent (13 versus 19) fewer MN on clothing-protected body sites than control children.

Results of multivariable regression analyses comparing logarithmically transformed incidence rates of MN in children from intervention and control centers remained significant (n = 504; *p* = 0.001) after adjustment for cluster sampling and the confounding effects of age at baseline, socioeconomic status of the suburb of residence (SES), and the median number of hours spent outside on a typical day. Similarly, results of multivariable regression analyses comparing the logarithmically transformed numbers of incident MN of children from intervention and control centers remained significant (n = 519; *p* = 0.020) after adjustment for cluster sampling and the confounding effects of age at baseline, SES, the number of grandparents of European descent, and the frequency of beach visits during the warmer months of the year.

The results of a J-T test conducted for the 334 children in the intervention group provided some evidence of a statistically significant linear relationship between an increasing total number of incident MN acquired by intervention children and decreased clothing compliance at the intervention center they attended across five ordered categories (J-T test = 23,234; standardized test statistic = 2.034; *p* = 0.042). Consequently, the hypothesis that the distribution of MN was the same across ordered categories of clothing compliance was rejected. The median total number of incident MN [IQR] was lowest at 8 MN [IQR 2,17] for children at intervention childcare centers where more than 98 percent of children on average were observed wearing study clothing outdoors. This compares to a median of 12 MN [IQR 6,21] for children at intervention centers where the average clothing compliance was >95–98 percent, 12 MN [IQR 5.5, 23] where clothing compliance was >90–95 percent, 15 MN [IQR 5.5, 24] where compliance was 80–90%, and 14.5 [IQR 6, 25.25] where clothing compliance was below 80%. Although this finding suggested that there was a dose–response relationship in terms of the total number of incident MN in intervention children (continuous variable) that varied with the average level of clothing compliance observed at these centers (ordinal variable), the Mann–Whitney U test post hoc pairwise comparisons were not statistically significant after using the Bonferroni correction to adjust for multiple comparisons. A further J-T test was conducted using the rate at which new MN were acquired per month of follow-up as the continuous variable. A statistically significant result was achieved for the five ordered categories of clothing compliance (J-T test = 23,384; standardized test statistic = 2.185; *p* = 0.029). The median rate at which MN were acquired was 0.33 MN/month [IQR 0.113, 0.550] for the highest (>98%) clothing compliance category; 0.46 MN/month [IQR 0.270, 0.796] for the >95–98% compliance category; 0.45 MN/month [IQR 0.244, 0.683] for the >90–95% compliance category; and 0.48 MN/month [IQR 0.251, 0.760] and 0.54 MN/month [IQR 0.313, 0.808] for the lowest category of compliance (<80%). The Mann–Whitney U test post hoc pairwise comparisons revealed that the rate of incident MN/month differed significantly for the highest (>98%) clothing compliance category compared to the lowest (<80%) clothing compliance category, even after adjusting for multiple comparisons (10 paired tests) using the Bonferroni correction (Mann–Whitney U test = 1021; standardized test statistic = 2.926; *p* = 0.017). The nine remaining pairwise comparisons were not statistically significant after applying the Bonferroni correction (comparison of the highest compliance subgroup and the second lowest compliance subgroup had the smallest adjusted one-sided *p*-value of the remaining nine comparisons at *p* = 0.055).

## 4. Discussion

The number of MN is a robust risk factor for cutaneous melanoma [5,17,18,19,20,21,22,24,25,32,39,41,42,43,45]. This, together with sharing risk factors with cutaneous melanoma, makes them appropriate surrogate endpoints for evaluating the efficacy of sun protection interventions [46,47,48]. Our c-RCT is the first to provide convincing evidence that the effect of sunlight on young children can be mitigated by regularly wearing UPF 30-50+ rated clothing that covers a considerable proportion (at least half) of the body. The major finding was that diligent use of sun-protective clothing was effective in preventing the development of a significant proportion of new MN in young children. Children who attended “intervention” childcare centers developed significantly fewer MN overall (whole body surface excluding the buttocks, genitals, and scalp) and on anatomical sites specifically protected by the study garments (i.e., the trunk, posterior neck, upper arms, and thighs) than children attending “control” centers (i.e., children who wore their own clothing to childcare for the duration of the study). Furthermore, the median incidence rate of MN acquired per month of follow-up was significantly higher for children attending “control” centers than for children attending “intervention” centers. The rate of proliferation of new MN also plateaued for intervention children who were retained in the study for longer periods, while in control children, MN incidence rates continued to rise with the duration of follow-up. Thus, the longer children spent in the study, the greater the difference in incident MN counts between the two groups. Thus, by 3.5 years of follow-up, children attending intervention centers had 24.3 percent fewer new MN overall and 31.6 percent fewer new MN on clothing-protected skin compared to control children. These findings are further supported by the significant inverse association between the rate of development of new MN among intervention children and the observed level of clothing compliance. This suggests that more MN were prevented in children who attended more compliant intervention centers compared to those attending less compliant centers.

A recent literature search identified eight sun protection intervention studies conducted in prepubescent children that used MN as a surrogate endpoint [47,48,62,63,64,65,66,67,68] (Table 4). All studies other than the “Kidskin Study” [47,48] were designed as RCTs. Six of these eight intervention studies aimed at preventing MN (Table 4) involved primary school-aged children [47,48,62,65,66,67,68], while one study recruited infants aged 0–6 months through primary health care settings in Denver, U.S.A. [64], and another recruited infants and preschool-aged children from childcare centers in Germany [63]. The primary focus of all of the studies reported in Table 4 was educating children (and in some cases, their parents) about sun protection [47,48,63,64,65,66,67,68] and/or providing them with sunscreen [62,63,64]. None of the intervention studies identified chose sun-protective clothing as their primary intervention strategy (Table 4), further emphasizing the uniqueness of our c-RCT.

To our knowledge, only one other RCT (a sunscreen RCT conducted in Canadian schoolchildren [62]) aside from our c-RCT has reported a clear-cut statistically significant reduction in total body incident MN count in children of both genders (Table 4). In addition, at the 6-year follow-up of the curriculum- and policy-based intervention study “Kidskin” conducted in Western Australian primary schools, a significant reduction in the number of incident MN on the posterior trunk was reported for boys [48]. Although clothing was not the primary focus of the “Kidskin Study”, children in the most intensive intervention group were offered low-cost sun-protective swimwear that covered the trunk [47,48]. A subsequent analysis of these data grouped for all children (ignoring assigned study group) indicated that 12-year-old children whose parents reported they always wore clothing covering their posterior trunk when outside and avoided spending time outdoors around solar noon had fewer MN on their backs than those who did not [69]. Their findings [48,69] are consistent with our earlier comparison of MN counts on the posterior trunk of 1–3-year-old NQ children from two different cohorts established before (1991) and after (1999–2002) swim-shirts became popular among Australian children [70]. Together, these studies [48,69,70] provide support for the findings of our c-RCT.

Although ours is still the only RCT conducted to date that has been specifically designed to quantify the proportion of MN that can be prevented by clothing, other researchers have examined the association between clothing and MN in cross-sectional studies of young children and have produced results consistent with the findings of our c-RCT [50,51,71,72]. For example, a cluster prevalence survey of 193 children attending childcare centers in southeast Queensland showed that children 1–3 years of age who wore legionnaire-style hats had lower MN counts than children who did not wear a hat [71]. Likewise, those children who always wore a hat also had fewer MN on the head and neck than those who rarely wore a hat [71]. Furthermore, a study conducted in 2–7-year-old children from Germany who wore more clothing when at the beach or an outdoor swimming pool had 24 percent fewer MN on their bodies than children who wore less clothing [72]. This is the same proportion of MN that we prevented in preschool-aged intervention children who remained in our clothing c-RCT for at least 3.5 years.

In 1996, Australia pioneered a relative ranking of the sun-protective capabilities of clothing based on the percentage of UVR transmitted through fabric, known as UPF [73], which led to the first industry standard for sun-protective clothing [54]. UPF swing tags were introduced as part of this widely adopted voluntary standard [74,75,76,77]. While most standards now specify minimum garment coverage requirements for sun-protective clothing [75,78,79], some, such as those in the U.S.A., do not [76,77]. The current European [75], Australian [78], and New Zealand standards [79] require upper-body garments to cover the trunk and at least three-quarters of the upper arm. However, the European standard requires lower-body garments to cover the area from the waist to just below the knees [75], while the current Australian and New Zealand standards only require them to cover from the waist to half-way between the crotch and the knees [78,79]. Similar BSA coverage to the minimum requirement specified in the current European standard [75] was sufficient to prevent the development of a statistically significant proportion of MN in our c-RCT. Given mounting evidence for the importance of sun-protective clothing in preventing sun-damaged skin, MN, and skin cancer, it may be timely to consider the pros and cons of making standards for the evaluation, classification, and labeling of sun-protective clothing regulatory (madatory) rather than voluntary.

Standards based only on the UPF of the fabric enable suppliers of brief swimwear and apparel to mislead consumers by displaying a UPF swing tag claiming a sun-protective advantage [80,81]. Clothing labels provide an opportunity to inform consumers about the importance of garment coverage in preventing skin cancer, particularly where children are concerned. Effort continues to be invested in developing a reliable and cost-effective test method for quantifying garment coverage which can then be reported together with the fabric UPF on the swing tags of sun-protective garments [80,81,82]. The ‘Garment Protection Factor’ (GPF), which combines both of these important metrics into a single rating, has been proposed to encourage manufacturers to design clothing that exceeds minimum garment coverage standards [80]. More informative labeling makes the identification of quality sun-protective garments easier for schools and sporting clubs responsible for providing uniforms for children in UVR-intense environments and reinforces the idea that simple alterations such as lowering hems and slightly increasing sleeve length can improve sun protection without incurring additional costs [83]. Likewise, influencing school uniform procurement policies to specify minimum garment coverage requirements in regions where uniforms are mandatory, as we have achieved for government schools across Queensland in conjunction with the Queensland Department of Education’s Procurement Branch, helps schools to meet their duty of care by making the acquisition of quality sun-protective school uniforms easier [84,85]. This model provides a simple and sustainable solution for the supply of sun-safe school uniforms, which could potentially be adapted for use in other regions [84,85]. Our c-RCT demonstrates that young children who diligently wear UPF 30-50+ clothing that covers about half their BSA develop significantly fewer MN both overall and on specifically protected anatomical sites compared to control children who wore their own clothes throughout the study. However, recent studies of sun protection policies in early childhood settings in Australia [86,87], New Zealand [88], and Germany [89] demonstrate that suitable clothing is currently the least used form of personal sun protection for children attending early childhood services and that hat use and sunscreen are the most common sun protection policy inclusions [86,87,88,89], corroborating findings from almost two decades ago [49]. Serial cross-sectional data collected in Australian early childhood settings in the decade prior to 2018 also report that a small and declining proportion of services used sun-protective clothing to protect young children from the sun [87]. Almost 62 percent of Australian children observed at an outdoor festival held in spring of 2015 in New South Wales wore a sun protective hat (wide-brimmed, bucket, or legionnaires hat), though most wore short sleeves, and just under five percent of children wore sun-safe sleeves (at least three-quarter-length) [90]. The proportion of Caucasian prepubescent children (approximately 0–12 years of age) observed at an annual outdoor motorsport event in NQ wearing sun-safe sleeves was also quite small (9.8 percent in 2009 [91] and 13.9 percent in 2022 [92]), while the proportion of children observed wearing a sun-safe hat at this event declined sharply from 45.1 percent in 2009 [91] to 26.4 percent in 2022 [92]. Interestingly, declining hat use in children does not appear to be a phenomenon that is just limited to Australia. The proportion of prepubescent children observed wearing a hat at outdoor theme parks in the U.S.A. declined from 40.2 percent in 2000 to 22.9 percent in 2007 (*p* < 0.001) and 15.4 percent in 2017 (*p* = 0.032) [93]. Statewide data collected in 2020 indicate that only 16 percent of children from Queensland use sunscreen daily [8], while data collected by the Cancer Council over the past two decades show that the use of well-accepted sun protection measures, including hats, sunglasses, sunscreen, and shade, have not consistently improved in the Australian population since 2003–2004 when data collection commenced [94]. Furthermore, the proportion of Australians who report wearing sun-protective pants decreased by 10 percent to 36 percent by 2016–2017, while sun-safe sleeve use only declined slightly to 17 percent over the same period [94]. These studies demonstrate the relevance of the findings of our unique c-RCT and, together with data showing that almost half of Queensland children still experience sunburn each year [8], signify an urgent need to raise awareness about the importance of increased clothing coverage for children.

Primary health care providers should stress the importance of garment coverage, particularly when recommending sun protection for children, and public health campaigns should reinforce this important message. As fashion has the potential to positively influence sun-protective behaviors [95], it is important to develop an alliance with the fashion industry [81,83], including those who train fashion designers and influence the curriculum. Introducing innovative awards which recognise excellence in sun-protective clothing design may help encourage a greater focus on designs with higher BSA coverage while maintaining their esthetic appeal for the relevant target group [95]. Designing fashionable, affordable, and sun-protective garments that are sufficiently comfortable enough for everyday use may increase their popularity and improve sun-protective behavior in high-risk populations [81].

While the reduction in the absolute number of MN in young children in the intervention group of our c-RCT was not large, the differences in total MN counts and clothing-protected body sites between groups were significant over the 3.5 years of follow-up and provide the best indication of the proportion of MN in young children that can be prevented. We commenced this study in November 1999 because clothing was underutilized as a means of protecting young children from the intense solar UVR environment of Queensland from overexposure to sunlight and the associated acquisition of MN. Although the data presented are fairly old, we have demonstrated that there have been no significant improvements in the personal sun protection practices of young Australian children since we commenced our c-RCT [8,49,86,87,90,91,92,94], and recent findings suggest that clothing is still vastly underutilized by children as a form of sun protection both in Australia [8,49,86,87,90,91,92,94] and abroad [88,89,93]. Thus, our findings are as relevant in 2023 as they were during the study period, and our study remains the only reported RCT that has measured the efficacy of clothing in reducing the incidence of pigmented moles in young Caucasian children. We chose childcare centers because we could access children and introduce protective clothing. By regularly wearing UPF 30-50+ clothing that covered about half of the BSA (i.e., the trunk, posterior neck, upper arms, and thighs), almost 32 percent of new MN were prevented on clothing-protected body sites compared to control children who wore their own clothes at childcare. This c-RCT was not designed to alter childcare center policy in Australia, but it demonstrates that all children, whether in care or at home, can achieve effective sun protection by wearing adequate clothing.

Other limitations of the study have been described elsewhere [46]. Briefly, they include the groups being slightly different at baseline; however, most of the assessed sun-protective behaviors favored control centers (i.e., their policies and behaviors appeared to be more sun-protective) such that the slight imbalances between intervention and control centers would result in a bias towards the null. Recruitment of children continued for 2.5 years because the target population was not sufficiently large for simultaneous recruitment, and attrition rates were considerable because the population of Townsville is relatively transient [46]. However, where possible, a final examination was undertaken before a child’s departure from a participating childcare center. Our frequent presence to conduct MN examinations may also have sensitized the control group to the purpose of the study, potentially biasing results towards the null hypothesis. Further follow-up of this cohort in early adulthood would be beneficial to verify whether this important surrogate endpoint for melanoma risk was modified in the longer term or whether the development of these MN was delayed rather than prevented by the intervention. Expanded access to systems providing total body photography with sequential digital dermatoscopic imaging support a growing teledermatology network in Australia, making follow-ups with this cohort a more viable proposition than ever before, even if a considerable proportion of them have relocated elsewhere in the country. Data linkage to Australia cancer registry data may also generate information of value when conducting follow-ups with this cohort.

## 5. Conclusions

Our RCT remains the first and only longitudinal study to demonstrate that it is possible to prevent the development of a significant proportion of pigmented moles in young Caucasian children by dressing them in UPF 30-50+ clothing that covers at least half of their body on a daily basis. Children attending intervention childcare centers developed almost 25 percent fewer MN overall and 32 percent fewer MN on clothing-protected body sites compared to children attending control centers. This should, by implication, reduce their risk of developing melanoma as adults. A statistically significant reduction in the development of new MN was achieved in this “real world” study involving interventions undertaken by parents and staff in childcare centers, which contrasts with the highly controlled experiments conducted under optimal circumstances by committed researchers. Study garments for everyday use were designed to be loose-fitting and cool to wear. The intervention was well tolerated by most children (including thermal comfort), with only three children refusing to wear the clothing (final MN count performed) during the 5.5-year study period. It was effective even though community awareness about sun protection was high due to decades of skin cancer prevention campaigns and despite the fact that hats and sunscreen were already worn by almost all Australian children attending formal childcare when the study began, making it more difficult to disprove the null hypothesis.

The benefits of this clothing intervention in terms of preventing a significant proportion of MN, the relative ease with which the clothing was adopted by childcare centers, and evidence demonstrating that sensible longer-length clothing is still vastly underutilized in early childhood settings, both within Australia and abroad, suggest that it may be prudent to conduct dissemination studies to investigate how the uptake of sun-protective clothing can be improved in young children. One option that may be worthy of consideration for intense UVR environments is trialing the introduction of sensibly designed, affordable sun-safe clothing similar to that used in our study (e.g., crew-neck, cool, loose-fitting T-shirts with elbow-length sleeves and knee-length shorts) as a mandatory uniform for children attending early childhood services.

In conclusion, increased clothing cover should be recommended for children exposed to high levels of UVR. All industry standards for sun-protective clothing should specify minimum garment coverage requirements in their definition of sun-protective clothing. This is particularly important for children’s clothing. Sun-protective clothing standards that only consider the UPF of the constituent fabric, such as those in the U.S.A., allow clothing that does not cover significant areas of skin, such as bikinis, to claim a sun-protective advantage or to be marketed as sun-protective (including on labels and packaging) if the fabric has a high UPF (e.g., 50+). Such information is misleading for the public, especially when trying to protect children. Sun-protective clothing standards should require testing and reporting of garment coverage on permanent labels, swing tags, and packaging either alongside the fabric’s UPF or integrated into the GPF, which is a single metric that takes both factors into account. This approach rewards manufacturers for designing and producing garments that cover more skin while also helping consumers to make a more informed choice when purchasing sun-protective clothing.

## Figures and Tables

**Figure 1 cancers-15-01762-f001:**
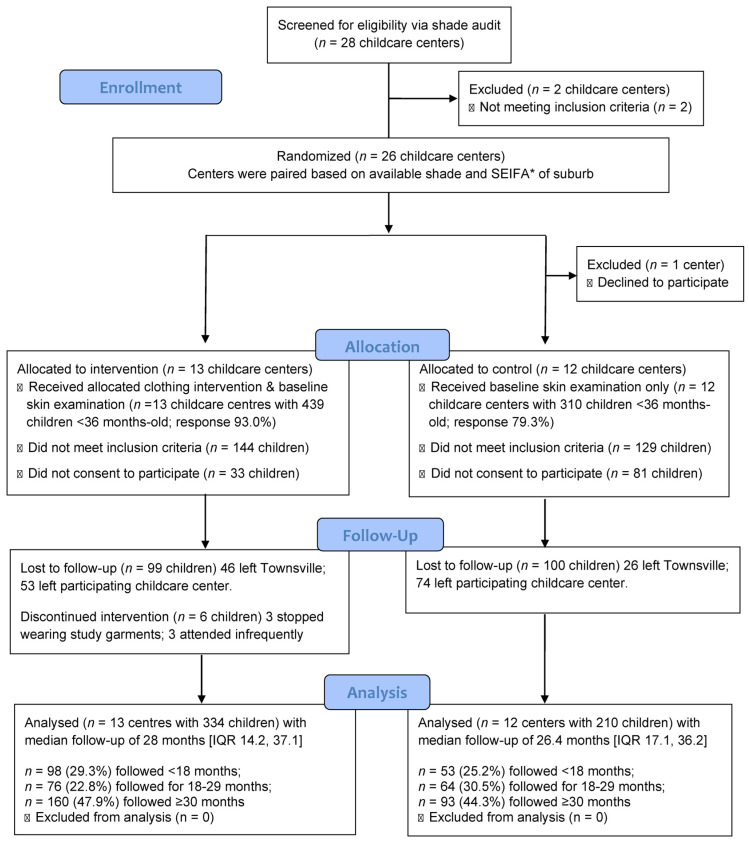
CONSORT 2010 flow diagram [57] showing 1–35-month-old children from Townsville, Australia, who were randomized by childcare center (clusters) to parallel arms of this cluster randomized controlled trial for up to 4 years. Abbreviations: SEIFA, socioeconomic index for areas [55]; *n* = number; < less than; ≥ greater than or equal to.

**Figure 2 cancers-15-01762-f002:**
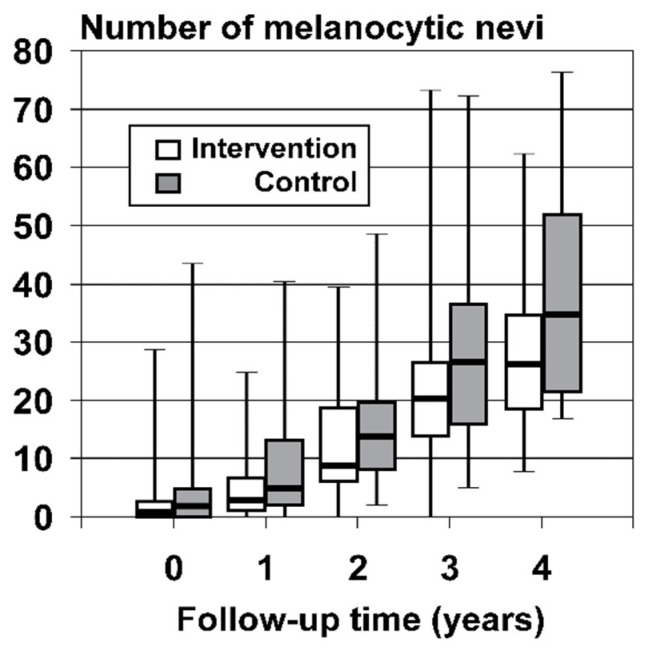
Total number of new melanocytic nevi developed by 1–35-month-old children participating in a cluster randomized controlled trial of sun-protective clothing in children attending a participating childcare center in Townsville, Australia.

**Table 1 cancers-15-01762-t001:** Baseline characteristics of the 25 childcare centers participating in the cluster randomized controlled trial in Townsville, Australia.

Characteristics	Intervention Childcare Center (*n* = 13)	Control Childcare Center (*n* = 12)	*p*-Value
% childcare centers under private management	61.5%	58.3%	*p* > 0.999
Median of max number of licensed places at the center (IQR)	75 (74, 75)	66 (46.5, 75)	*p* = 0.17
% centers with a sun protection policy	100%	100%	*p* > 0.999
% “SunSmart” centers	7.7%	8.3%	*p* > 0.999
% centers in suburbs with moderate socioeconomic indicators *	61.5%	75.0%	*p* = 0.34
Mean measured protection provided by shade structures ** (SD)	54.2 (19.2)	57.3 (14.3)	*p* = 0.67
% centers offering moderate to good shade for outside areas	61.5%	50.0%	*p* = 0.60
**Characteristics**	**Intervention Baby Units (n = 44) #**	**Control Baby Units (n = 32) #**	**p-Value**
% units that applied sunscreen routinely to all children	68.2%	68.8%	*p* = 0.97
% units that provided sunscreen for children (of the 64 units who applied sunscreen)	47.2%	60.7%	*p* = 0.53
% units that applied sunscreen to:			
Face	81.8%	84.4%	*p* = 0.84
Anterior Neck	36.4%	65.6%	*p* = 0.14
Forearms	81.8%	87.5%	*p* = 0.66
Lower Legs	68.2%	68.8%	*p* = 0.97
All four sun-exposed body sites	31.8%	46.9%	*p* = 0.36
Median hrs spent outside per day (IQR)	3.1 (1.9, 4.4)	2.7 (2.0, 5.0)	*p* = 0.57 *p* = 0.96§
% units where children spent time outside with their back exposed to the sun (duration per day outside)	4.5% (max 1 h/day)	9.4% (max 40 min/day)	*p* = 0.54

^#^ Analysis across 44 intervention and 32 control baby and toddler units grouping children according to age between birth and 3 years of age; childcare centers had 24 units to care for children under 3 years old; analysis adjusted for the cluster effect of childcare centers; * socioeconomic categories were based on SEIFA indicators [55]; ** shade audit (the lower the score the higher the protection). Abbreviations: max, maximum; IQR, interquartile range; hrs, hours; mins, minutes; SEIFA, socioeconomic index for areas.

**Table 2 cancers-15-01762-t002:** Baseline characteristics of 544 Australian children (1–35 months-old) attending 25 childcare centers in Townsville, Australia, who had a follow-up examination for melanocytic nevi (MN). Results were adjusted for cluster sampling.

Phenotypic Characteristics of Children	Total (*n* = 544)	Intervention (*n* = 334)	Control (*n* = 210)	*p*-Value
Mean age [months] (±SD)	15.4 (6.5)	14.8 (6.3)	16.3 (6.7)	*p* = 0.222
% girls	46.0%	44.9%	47.6%	*p* = 0.556
% fair-skinned children	97.2%	97.0%	97.6%	*p* = 0.659
% children with blue or green eyes	66.7%	65.9%	68.1%	*p* = 0.787
% children with fair or blond hair	66.7%	63.5%	71.9%	*p* = 0.167
% skin ”always burns” after 30 min sun exposure ¶	33.7%	33.9%	33.3%	*p* = 0.702
% skin ”never tans” after 30 min sun exposure ¶	14.9%	14.1%	16.3%	*p* = 0.694
Mean % skin reflectance of the inner upper arm (±SD)	72.1 (2.5)	72.2 (2.5)	72.1 (2.5)	*p* = 0.597
% children with 4 Caucasian grandparents †	83.0%	83.9%	81.6%	*p* = 0.209
**Demographic characteristics of children**				
% children with 1+ parents with a degree †	36.8%	39.2%	32.8%	*p* = 0.504
% children living in low-SEIFA suburb ‡	28.9%	24.9%	35.2%	*p* = 0.200
% children born in the tropics †	91.7%	92.6%	90.3%	*p* = 0.579
**Sun exposure of children**				
Median hours spent outside on a typical day [IQR] †	2.3 [1.1,3.9]	2.3 [1.1, 3.9]	2.5 [1.3, 4.0]	*p* = 0.980 *p* = 0.817 §
Median total hours spent outside in warmer half of the previous year [IQR] †	178 [31, 484]	178 [27, 465]	192 [38, 521]	*p* = 0.431 *p* = 0.630 §
Median number of hours spent playing in water on a typical day in warmer half of the previous year [IQR] †	2 [1, 3.5]	2 [1, 3.5]	2 [1, 3.5]	*p* = 0.785 *p* = 0.810 §
% children who swim in an outdoor pool 1+/week during the warmer months †	41.2%	39.0%	44.9%	*p* = 0.378
% children who visit the beach 1+/week during the warmer months †	8.5%	9.9%	6.1%	*p* = 0.596
% children who played outside “almost every day” during the warmer months †	44.5%	41.2%	50.0%	*p* = 0.299
**Sunburn characteristics**	**Total (*n* = 544)**	**Intervention (*n* = 334)**	**Control (*n* = 210)**	***p*-value**
% children with at least one sunburn with “redness without peeling” †	43.8%	42.4%	46.2%	*p* = 0.404
% children with at least one “peeling sunburn” †	3.9%	3.4%	4.6%	*p* = 0.528
% children with at least one sunburn that was “very painful with blistering” †	0.6%	0%	1.5%	*p* = 0.022
% children with a sunburn severity score of 2+ †	4.2%	3.4%	5.6%	*p* = 0.403
Median extent of sunburn score [IQR] †	0 [0, 0.13]	0 [0, 0.13]	0.05 [0, 0.14]	*p* = 0.222 *p* = 0.184 §
Median sunburn history score weighted according to extent and severity [IQR] †	0 [0, 0.13]	0 [0, 0.13]	0.05 [0,0.14]	*p* = 0.220 *p* = 0.128 §
**Use of sun protection**				
% parents who used sunscreen on child at home †	91.3%	90.1%	93.4%	*p* = 0.288
% children who use SPF 30+ sunscreen ∫	76.8%	77.8%	75.3%	*p* = 0.600
% parents who “almost always” apply sunscreen to their child at home in summer †	49.4%	49.7%	49.0%	*p* = 0.527
% parents who “almost always” apply sunscreen to their child at home in winter †	30.8%	30.3%	31.6%	*p* = 0.838
Mean number of body sites to which sunscreen was usually applied (±SD) †	7.5 (3.4)	7.3 (3.5)	7.8 (3.2)	*p* = 0.062
Median BSA to which sunscreen was usually applied [IQR]†	0.44 [0.3, 0.5]	0.44 [0.3, 0.5]	0.45 [0.3, 0.5]	*p* = 0.109 * *p* = 0.068 §
Median sunscreen score weighted by BSA and frequency of use [IQR] †	2.1 [0.9, 3.4]	2.0 [0.8, 3.4]	2.1 [1.0, 3.4]	*p* = 0.397 * *p* = 0.162 §
Median hat use score weighted by hat style and frequency of use [IQR] †	16 [8, 24]	16 [9, 24]	16 [8, 24]	*p* = 0.049 * *p* = 0.143§
% children who usually wear Lycra suit in water †	64.8%	66.0%	62.8%	*p* = 0.302
Median swimwear protection score weighted by style and frequency of use [IQR] †	25 [20, 25]	25 [20, 25]	25 [15, 25]	*p* = 0.259 * *p* = 0.091§
**Outcome**				
Median total melanocytic nevus count [IQR]	1 [0, 4]; range 0–44	1 [0, 3]; range 0–29	2 [0, 5]; range 0–44	*p* = 0.028 * *p* = 0.183§

¶ A total of 75 parents answered “did not know” in response to this question because their child was too young for them to answer with certainty; † questionnaire responses missing for 25 children; ∫ 117 parents neglected to write the SPF of the sunscreen they applied most often to their child; ‡ socioeconomic status based on SEIFA indicators [55]. * first *p*-value relates to non-parametric test comparing median values without cluster adjustment; § second *p*-value relates to comparison of mean values of log-transformed variables adjusted for cluster sampling. Abbreviations: SD, standard deviation; IQR, interquartile range; SPF, sun protection factor; min, minutes; SEIFA, socioeconomic index for areas; BSA, body surface area.

**Table 3 cancers-15-01762-t003:** Comparison between intervention and control children with regards to (i) total number of new melanocytic nevi (MN); (ii) number of new MN acquired per month; and (iii) number of new MN on body sites protected by study garments ¶ by duration of follow-up. Results based on 544 children followed for up to 4 years.

Duration of Follow-Up	Intervention Arm (*n* = 334 Children)	Control Arm (*n* = 210 Children)	*n* *p*-Values
Median counts of incident MN present at final assessment [IQR] by duration of follow-up			
<18 months	3 [1, 7]; range 0–25	5 [2, 13.5]; range 0–41	*n* = 151
18–29 months	9 [6, 19]; range 0–40	14 [8.25, 20]; range 2–49	*n* = 140
30–41 months	20.5 [14, 27]; range 0–74	27 [16, 37]; range 5–73	*n* = 207
≥42 months	26.5 [18.75, 35]; range 8–63	35 [21.5, 52.25]; range 17–77	*n* = 46
Overall	12.5 [5.75, 23]; range 0–74	16 [8, 30]; range 0–77	*p* < 0.001;
			*p* = 0.011;
			*p* = 0.020§
Median incidence rate [IQR] (i.e., new MN acquired per month) by duration of follow-up			
<18 months	0.25 [0.13, 0.55]	0.46 [0.15, 1.13]	*n* = 151
18–29 months	0.38 [0.25, 0.70]	0.54 [0.35, 0.83]	*n* = 140
30–41 months	0.58 [0.37, 0.79]	0.75 [0.50, 1.13]	*n* = 207
≥42 months	0.57 [0.43, 0.77]	0.81 [0.50, 1.16]	*n* = 46
Overall	0.46 [0.25, 0.71]	0.68 [0.37, 1.04]	*p* < 0.001;
			*p* < 0.001;
			*p* = 0.001§
Median counts of new MN at sun-protected body sites [IQR] by duration of follow-up			
<18 months	1 [0.75, 3]; range 0–14	3 [0.5, 6.5]; range 0–16	*n* = 151
18–29 months	5 [2, 9]; range 0–22	6 [3.25, 11]; range 1–27	*n* = 140
30–41 months	8.5 [6, 14]; range 0–44	12 [8, 20]; range 0–41	*n* = 207
≥42 months	13 [10.25, 18]; range 2–32	19 [12, 26.25]; range 6–31	*n* = 46
Overall	6 [2, 12]; range 0–44	8 [4, 15.25]; range 0–41	*p* < 0.001;
			*p* = 0.027;
			*p* = 0.021§

¶ body-sites specifically protected by study garments included the posterior neck, the trunk, the upper arms and the thighs. IQR, interquartile range. §: the first *p*-value relates to non-parametric tests comparing median values without cluster sampling adjustment, the second *p*-value relates to comparisons of the mean values of log-transformed variables adjusted for cluster sampling, and the third *p*-value relates to results of multivariable regression analysis adjusted for the confounding effects of (1) age at baseline, socioeconomic indicator of the suburb where the child lived, the number of grandparents of European descent, and the frequency of being at the beach during the warmer months of the year for incident number of MN (n = 519); (2) age at baseline, socioeconomic indicator of the suburb where the child lived, and the median number of hours spent outside on a typical day for incidence rate (n = 504); and (3) age at baseline and the frequency of being at the beach during the warmer months of the year for the median counts of incident MN at sun-protected body sites (n = 519).

**Table 4 cancers-15-01762-t004:** Summary of the design and outcome of intervention studies published before December 2022 that aimed to reduce the number of melanocytic nevi (MN) developing in prepubescent children.

Study Name, Authors, and Study Date	Study Population at Baseline	*n*	Design, Sun Protection Strategies Used in Intervention, and Level of Randomization	MN Exam Methods	Outcome
Vancouver Mole Cohort Study 1993–1996 Gallagher et al. 2000 [62]	Grade 1 (6–7 yrs) and 4 (9–10 yrs) Caucasian children in British Columbia	Baseline (n = 483): controls (n = 236), S/S group (n = 222) 3 yr follow-up (n = 309)	RCT of S/S (children individually randomized) Treatment group supplied with SPF30 S/S and directions for use when the child was to be in the sun for at least 30 min. **Main Strategy ¶**	Whole-body MN counts (all sizes) by dermatologist excl. buttocks, genitals, and scalp. Breast not examined in girls.	S/S group developed significantly fewer MN than control children (median 24 vs. 28; *p* = 0.048). Freckled S/S group developed fewer MN than freckled controls.
Kidskin Study 1995–1999 Milne et al. 2002 [47] English et al. 2005 [48]	Grade 1 (5–6 yrs) Caucasian children in Perth, Australia	Baseline (n = 1615) 4-year follow-up (n = 1453) 6-year follow-up in 2001 (n = 1116) comprised of 14 control schools (n = 484), 11 moderate intervention schools (n = 354), and 8 high intervention schools (n = 278)	Non-random cluster trial with 3 arms. School curriculum and policy-based sun protection intervention. Specially designed 4-year sun- protection curriculum and given guidelines on school sun protection policies. “High” group also received program materials in summer vacations and were offered discounted sun-protective swimwear. **Main Strategies † §**	Partial-body MN counts (all sizes) from projected image by lay examiners. MN counted on face, arms, and back (excl. shoulders) for both genders. MN on chest also counted in boys. MN on lower limbs not counted.	Mean new MN on back at follow-up: controls = 6.6 MN; moderate group = 5.2 MN; high group = 5.3 MN (*p* = 0.09; NS). Hat use improved in the high group but NS difference in MN on face and arms. Weak evidence of reduced counts of new MN. Mean new MN on boys’ backs at follow-up: controls = 7.9 MN; moderate group = 5.9 MN; high group = 6.4 MN; *p* = 0.0009. MN by gender: NS.
1998–2001 Bauer et al. 2005 [63]	2–7 year-olds at 78 public nursery schools (childcare centers) in Stuttgart and Bochum, Germany	Baseline (1998) n = 1812; 3-year follow-up n = 1232: control group (n = 398), education group (n = 369), S/S and education group (n = 465)	Cluster-RCT (3 arms). Moderate intervention group parents sent sun protection education letter 3x/year; high intervention group sent sun protection education and a free bottle of SPF25 S/S per year. **Main Strategies ∫ ¶**	Whole-body MN (all sizes) counted by dermatologist.	NS difference in MN counts between groups. Median new MN [IQR]: controls = 8 [4,14]; education group = 8 [4,14]; education + S/S group =9 [6, 14]; *p* = 0.101 (NS).
1998–2001 Crane et al. 2006 [64]	0–6 month-olds recruited from 14 primary care practices servicing ~29% insured population in Denver- –Boulder area, USA	728 infants and their parents 3-year follow-up of MN	Cluster-RCT. Sun protection advice provided to parents by healthcare providers at each well-child visit from 2–36 months old. Child sun hat provided at first visit; 2 small S/S samples provided each visit from 6 months old; sunglasses provided at 12 months old; parent–child activities to teach about sun protection provided at child’s 3-year visit. **Main Strategies ∫ ¶ ‡ ***	Whole-body MN counts (≥2 mm only) by dermatologist or pediatrician.	NS difference in MN counts: control = 5.64 MN, intervention = 6.3 MN, *p* = 0.56 (NS).
SoleSi SoleNo-GISED 2001–2004 Naldi et al. 2007 [65]	Grade 3–4 ~8-year-old children at 122 primary schools in Italy	Baseline (n = 4921 with MN counts) Follow-up 2005 (n = 3933) comprised of control group (n = 1661) and intervention group (n = 2272)	Cluster RCT. School-based intervention designed to reduce sunburn rates through use of a curriculum (median 6 hrs including video) and distribution of educational booklets on sun protection to children and parents. **Main Strategies † ∫**	MN counted on upper limbs in subsample of 4921 children (44% of baseline sample).	NS impact of educational program on sunburn episodes or MN 1 year after intervention. Geometric MN at follow-up: controls, 6.4 MN; intervention group, 6.8 (NS difference).
Colorado Kids Sun Care Program 2004–2007 Crane et al. 2012 [66]	6 year-olds (born 1998) recruited from pediatric offices and community settings in Denver, USA	Baseline 2003–2004 Follow-up of MN n = 677 white non-Hispanic children annually in 2005, 2006, and 2007	RCT postal intervention. Educational newsletters posted to parents and children. Parent-reported use of S/S, protective clothing, hats, shade-seeking, and midday sun avoidance. **Main Strategy ∫**	Whole-body MN counts (all sizes) by a team of 4–7 healthcare providers excl. buttocks, genitals, and scalp.	NS difference in MN < 2 mm. NS effect for presence of MN ≥ 2 mm (*p* = 0.09), with the intervention group having fewer large moles in 2006 only but not at the other 2 follow-ups.
Sun Protection for Florida’s Children (SPF) project 2006–2008 Roetzheim et al. 2011 [67]	Grade 4 primary children at 24 schools in Florida, USA	Baseline 2006–7 24 Florida schools Control group (n = 239 children) Intervention group (n = 200)	Cluster RCT of a school-based educational intervention focused on increasing hat use. Education session delivered in schools by researchers (2x hour sessions/yr) and parents given 2 wide-brimmed hats per child (for home and school). **Main Strategies † ‡**	MN counted on head and neck by research assistants in a convenience subsample of 439 children.	Hat wearing at school improved from 2% (baseline) to 41% at 1-year follow-up and 19% at 2-year follow-up. NS difference between intervention and control groups wrt MN counts, tanning, and self-reported hat use outside of school.
RISC-UV project 2007–2009 (Tête Brûlée study) Mahé et al. 2011 [68]	Children at 52 Primary schools in greater Paris area	Baseline n = 828 with MN counts conducted 2007 Follow-up MN in 2009 n = 660 (mean age 10.8 yrs; 1:1 males:females))	Cluster RCT of a school-based educational intervention evaluated by administering questionnaires to the player in 6 soccer teams about the sun protection they used during and between matches at a spring 2009 tournament. **Main Strategies †**	MN counted on the arms and back by 2 trained nurses (≤2 mm, >2 mm) in 2007 and again in 2009.	Sun protection use by soccer players and public inadequate. Total MN and new MN acquired after 2 years of study were higher in the 344 children who practiced outdoor sports.

Key to strategies: ¶, free sunscreen provided; †, school curriculum-based intervention; ∫, informal education focus (including parental education); ‡, provision of free sun-protective hat(s); *, provision of free sunglasses; §, discounted sun-protective swimwear. Abbreviations: yrs, years; MN, melanocytic nevi; NS, non-significant; S/S, sunscreen; IQR, interquartile range; SPF, sun protection factor; wrt, with respect to.

## Data Availability

The data presented in this study are available on request from the corresponding author. The data are not publicly available due to privacy and ethical reasons.

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
