# Peer review of "Sun-Protective Clothing Worn Regularly during Early Childhood Reduces the Number of New Melanocytic Nevi: The North Queensland Sun-Safe Clothing Cluster Randomized Controlled Trial"

_cancers, 2023, doi:10.3390/cancers15061762_

Round 1

Reviewer 1 Report

The manuscript reports results from a cluster-randomized trial in Australian childccare centers addressing the question whether wearing sun-protective clothing during early childhood has an effect on nevi development. The research question is addressed in a longitudinal study with sufficient follow-up and repeated outcome measurements in a standardized way. From a methodologic point of view, this interesting study leads to meaningful data that really can answer the research question.

Main comment:

- The study has been conducted more than fifteen years ago. The paper describing design and baseline results of the study - which I even remember having read in the AJE when it appeared - has been published in 2005. It is highly unusual to read about the main findings of such a trial more than fifteen years later. The authors touch upon this issue in the Discussion, but they need to elaborate on it. For example, if the sun protection practice in Australian childcare centers has changed over the last decade, the results of the study would have no direct policy implications. The authors need to explain conclusively why the results of their old study are still important (there is no need to explain why an earlier publication was not possible).

Other comments:

- The Introduction provides a good motivation for the study. However, it is noticeable that many of the references given in this section are quite old and only very few more recent publications are cited. I missed, for example, the JAAD review by Scope et al (2016, see doi: 10.1016/j.jaad.2016.03.027) and the paper by Satagopan et al. (2015, see doi: 10.1016/j.annepidem.2015.05.004) to name just a few. Last year I reviewed for a special issue “Children and UV radiation” of another MDPI journal (Children) that compiled some recent studies in Europe. The editorial of this special issue provides an overview of the topic (see doi: 10.3390/children9040537). Readers of the Introduction would benefit from an update of the references in the Introduction, pointing them not only to the traditional but also to more recent work in this area of research.

- Did the study gather information in the childcare centers of the intervention group that enable defining a measure of compliance to the intervention elements? It would be interesting to see whether there is a difference in the outcome variables between very compliant and less compliant childcare centers and such an additional analysis - given that the nevi development is slower in the very compliant compared to the less compliant childcare centers - would strengthen the authors' conclusion.

- Table 4 is not referenced in the text. No explanation is given how the information compiled in Table 4 has been collected. The most recent study included in the overview has been published in 2012 which makes me wonder whether the literature search leading to the Table 4 has been performed recently or long ago. Not to be misunderstood: I found the information in Table 4 valuable and it fits to the topic, but the reader needs an explanation how and when the information has been collected.

- Table 3 provides the information about the difference between intervention and control group with respect to three (related) outcome variables. In the legend of Table 3 it is described that for the different outcome variables different sets of confounding variables have been used in the analysis. In the paragraph devoted to the statistical analysis on page 6 this has not been mentioned explicitly. Please explain the rationale.

- In Table 1, three p-values are given as "p=1". From a statistical point of view, p-values in finite samples can never be "1" (or "0"). In practice, statistical software provides rounded results for the p-values, which tempts authors to include such impossible results in their tables. This can be remedied by specifying "p>0.999".

- The section Material and Methods covers more than four pages and provides many relevant details of the study. Readability of this section would benefit from structuring the text into subsections for the different aspects described.

- The term ‘biomarker’ used for melanocytic nevi (l. 401) is quite unusual. In the context it is used “surrogate endpoint” would better fit.

Typo:  l. 179: sizes (instead of sites)

Author Response

This reviewer is thanked for their insightful comments which we believed have improved the clarity and impact of our revised manuscript.

Please see the attachment for our responses to reviewer One.

Reviewer 2 Report

Sun-protective clothing worn regularly during early childhood reduces melanoma risk: the North Queensland Sun-Safe Clothing cluster randomized controlled trial

The manuscript reports on an important study and I commend the authors for persevering to write up the data after so many years have passed since data collection was done. This is a well thought out study and the manuscript is exempt of errors / typos etc., except for one in line 322, and check for % versus percent in words. This manuscript was a joy to read.

I have few high-level comments for the authors to consider:

1It would be helpful if as early in the study description as possible the status quo of sun protection use/education was described for the schools at the time. It becomes clearer as the manuscript goes on, but it is not stated upfront that all schools had sun protection policies and all children wore hats, for example, as the original status quo. A slow read through reveals this information but a little late. For example, in the abstract, line 32, ‘remained unchanged’ – meaning what? what was the situation that did not change?

2Please explain whether the intervention clothing was cool. My children hate long-sleeve swimsuits in the heat, even when swimming. It reads that children generally accepted the intervention clothing – did this include thermal comfort? It reads that the clothing provided in the intervention was not available on the market at the time of the study – was it available after? Otherwise what’s the use of knowing what works, yet it’s unavailable? Lastly, was the control group provided with the intervention clothing post-study (as an ethical responsibility)?

3Were the childcare centres private or government? This is really important to understand for other country scenarios. Please state upfront.

4Please check that the abbreviation BSA is written in full at first use.

5Line 178 – what does ‘while ever’ mean? Please rephrase.

6It was very clever to do the laundry of the sun protection garments as a mechanism to check numbers / use.

7Methods are very well described.

8Table 1 – example of saying there was a policy in place but not explaining what that policy required.

9Table 2 is very long and splits across pages – suggest making into Table 2a and Table 2b.

1Line 340 – fair-skinned sounds like taxidermy, please rephrase to children with fair skin.

1Figure 2 is very revealing. All along I thought that the intervention group got no nevi (of course this is wrong, but it was my perception) but the figure shows that the intervention group actually got quite a lot (n = 26) versus the control (n = 35) – perhaps make it a little clearer that this was the case; there were less melanocytic nevi, but still nevi present among intervention children.

12.   Excellent thinking in the study limitations.

Author Response

The reviewer is thanked for their suggestions and supportive comments.

Reviewer 3 Report

This manuscript presents results from an analysis of a cluster-randomized intervention study conducted in Townsville, Queensland, Australia between 1999 and 2005 that was designed to evaluate the impact of use of sun protective clothing to reduce development of melanocytic nevi in children. Recruitment was through child care centers for infants/children <36 months whose parents consented and completed baseline surveys. Nevi were assessed at baseline and annually for both groups for up to 3 years or loss of follow-up. The investigators recruited infants from child care centers and provided sunsafe clothing for use during the time the infant spent in the child care center as well as additional clothing items for use while the child was at home.

Primary comments:

1.       The title of this manuscript currently is “ Sun-Protective Clothing Worn Regularly During Early Childhood Reduces Melanoma Risk…”. However, the intervention focused on the impact of the intervention on the development of MN within a couple of years for very young children. Certainly, the rationale linking the development of new MN and melanoma is logical and probably true, but the citations in the introduction discuss the link between sun exposure and MN development in older children and other citations show a link between MN and melanoma in adults. No studies connect MN development in children and their actual risk of melanoma later in life. I suggest that the title be changed to better reflect this specific study…something like ‘Sun protective clothing worn regularly during early childhood reduces number of MN….’

a.       Relevant to this comment, is there potential to actually link these infant participants in 1999 to cancer incidence data for Australia?

22. This is an important study whose findings can have substantial relevance to the messages that new parents can receive about their infants and sun exposure. It is unfortunate that this is the first report of the study findings. A rationale was given (life intervenes), but these are important results that demonstrate the impact of sunsafe clothing for infants during the first three years for the development of MN.

a.       Children were evaluated for development of MN over the time period and appropriate care to reliability of the examinations was described.

b.       Rationale for why a cluster-randomized design was used was described and appropriate.

c.       Statistical methods were appropriate and group comparisons and adjustment for the clustered design were both considered. Figures and tables are all appropriate.

33. The description of the childcare centers (Table 1) was useful for readers to see that the groups were similar in care and sun protective behaviors at that point in time.  Of interest might be if any of the intervention childcare centers changed their policies/practices during the follow-up period. Are there any data to suggest any changes in childcare center behaviors after the three years?

44. The authors mention that they monitored the proportion of children wearing study-garments at intervention centers twice weekly. No mention of the ‘adherence’ to this use of garments was described and would be appropriate. Is such information available?

55. Appears that infants were not followed after leaving a childcare center and less than 10% of children remained in the child care center over the entire potential follow-up period. Something to consider for future studies as it would be relevant if parents with an infant in the intervention group continued with the coverage of their child.

Round 2

Reviewer 1 Report

The authors have addressed my remarks (and those of the other reviewers) appropriately and have made changes and additions during the revision that improved the presentation. It was a pleasure reading the revised manuscript and the rebuttal letter.

A final minor suggestion: the authors may consider to replace their analysis addressing differences between less compliant and more compliant childcare centers of the intervention group by a more powerful statistical analysis. The authors arbitrarily dichotomized the intervention group into two roughly equally-sized subgroups according to their compliance level and compared the nevi distributions between the subgroup using a nonparametric two-sample test. Alternatively, the authors could use their ranking of childcare centers according to their average clothing compliance (described in line 386ff) to define more than two (up to 13) subgroups that have an ordinal structure showing their clothing compliance (of course, if there were only negligible differences between childcare centers with respect to their average clothing compliance, it is not reasonable to define them as separate subgroups, thus, the number of subgroups will probably be lower than 13, but higher than two). The nonparametric Jonckheere-Terpstra test can then be used to test for an ordered alternative hypothesis within these independent subgroups. If there is a dose-response relationship between the level of clothing compliance and the intervention effect, the approach outlined above has a higher statistical power to detect it even in a medium-sized study sample. The analysis done by the authors revealed some evidence that children in more compliant childcare centers of the intervention group developed fewer new nevi than those in less compliant centers, but the observed effect missed statistical significance which is discussed by the authors as a limitation. My suggestion above intends to help the authors to use their data more efficiently, perhaps yielding a stronger conclusion.
